**Data Availability Statement:** All relevant data are publicly available and the accession numbers are provided within the paper and its Supporting information files.

# Non-invasive skin sampling detects systemically administered drugs in humans

**Morgan Panitchpakdi**[1,2], **Kelly C. Weldon**[1,2,3], **Alan K. Jarmusch**[1,2,4], **Emily C. Gentry**[1,2], **Arianna Choi**[1], **Yadira Sepulveda**[1], **Shaden Aguirre**[1,2], **Kunyang Sun**[1,2], **Jeremiah D. Momper**[1], **Pieter C. Dorrestein**[1,2,5], **Shirley M. Tsunoda**[1]*

**1** Skaggs School of Pharmacy and Pharmaceutical Sciences, University of California, San Diego, La Jolla, California, United States of America, **2** Collaborative Mass Spectrometry Innovation Center, University of California, San Diego, La Jolla, California, United States of America, **3** Center for Microbiome Innovation, University of California, San Diego, La Jolla, California, United States of America, **4** Immunity, Inflammation, and Disease Laboratory, Division of Intramural Research, National Institute of Environmental Health Sciences, National Institutes of Health, Research Triangle Park, North Carolina, United States of America, **5** Department of Pediatrics, School of Medicine, University of California, San Diego, La Jolla, California, United States of America

* smtsunoda@health.ucsd.edu

## Abstract

Clinical testing typically relies on invasive blood draws and biopsies. Alternative methods of sample collection are continually being developed to improve patient experience; swabbing the skin is one of the least invasive sampling methods possible. To show that skin swabs in combination with untargeted mass spectrometry (metabolomics) can be used for non-invasive monitoring of an oral drug, we report the kinetics and metabolism of diphenhydramine in healthy volunteers (n = 10) over the course of 24 hours in blood and three regions of the skin. Diphenhydramine and its metabolites were observed on the skin after peak plasma levels, varying by compound and skin location, and is an illustrative example of how systemically administered molecules can be detected on the skin surface. The observation of diphenhydramine directly from the skin supports the hypothesis that both parent drug and metabolites can be qualitatively measured from a simple non-invasive swab of the skin surface. The mechanism of the drug and metabolites pathway to the skin's surface remains unknown.

## Introduction

The skin, our largest organ accounting for 15% of total body weight, offers protection from the outside world while concurrently supporting immunological function and maintaining homeostasis. The skin's composition is complex and includes aqueous (*e.g.* sweat) and lipophilic (*e.g.* sebum) elements, enzymes, transporters, and microbes. The bacteria, viruses, and fungi comprising the skin microbiome function to protect against pathogenic organisms, assist the immune system, and breakdown chemicals [1]. While skin offers protection from the outside world, passage of chemicals are necessary for proper function. The relationship between blood and skin and the transference of chemicals between them is incompletely described.

**Funding:** This research was supported in part by the National Institutes of Health, UL1TR001442 (Tsunoda), UCSD Academic Senate Grant (Tsunoda), Intramural Research Program of National Institute of Environmental Health Sciences of the NIH (ES103363-01, Jarmusch), and NIH R01 GM107550 (Dorrestein). The funders had no role in study design, data collection and analysis, decision to publish, or preparation of the manuscript.

**Competing interests:** P.C.D is a scientific advisor to Sirenas, Galileo and Cybele and co-founder and scientific advisor to Ometa Labs and Enveda with approval by the University of California San Diego. M.P. is a research consultant to Ometa Labs.

Clinical measurement of biomarkers and drug concentrations typically rely upon invasive tests such as blood draws and tissue biopsies that are inconvenient for patients and can cause medical complications. Patients taking drugs with narrow therapeutic ranges require regular monitoring of plasma drug concentrations in order to maintain efficacy and prevent toxicity. While some forms of skin monitoring exist, for example, sweat testing for cystic fibrosis [2], sweat patches such as PharmCheck [3] and hydrogel micro patches [4–6] have been utilized, these are specific to testing topical, hydrophilic substances secreted with sweat.

Knowledge of xenobiotic transport through the skin stems from formulations which bring drugs from the skin to the systemic circulation, for example with the transdermal scopolamine patch for motion sickness. After release from the formulation, drug partitions into and diffuses through the stratum corneum to intercellular lipids. It then makes its way to the epidermis where it diffuses across viable layers into the dermis where it is absorbed into the systemic circulation via the capillary vessels [7]. Conversely, for many topical formulations of drugs there is extensive effort to avoid systemic exposure to minimize toxic effects of the compounds, for example with corticosteroid topical formulations. However, even with widely used products, the pathway from skin to blood is unclear. A recent study showed that ingredients in sunscreen were found in the systemic circulation at concentrations higher than the FDA safety threshold 23 hours post-application. Two of the sunscreen ingredients remained above the plasma threshold concentration for 21 days [8]. While variable absorption through the skin may be the rate-limiting step, the mechanisms and pathways from skin to blood and blood to skin need further exploration.

Systemically administered compounds (e.g. caffeine, citalopram) have been detected previously from skin swabs using untargeted mass spectrometry [9–11]. Orally administered drugs, such as sulfamethoxazole, were detected in non-invasive skin swab samples of kidney transplant patients [10]. A recent study used the sweat from individuals' fingertips to detect changes in caffeine over time after consumption of coffee [12]. Additional studies have demonstrated individuals fingerprints as a site of detection for a wide range of chemical compounds [13–19]. Another study measured caffeine in interstitial fluid (ISF) from the skin. In contrast to our study, caffeine in the ISF of the skin mirrored the concentration changes of caffeine in blood over time [20]. Sampling from the ISF involves puncturing of the skin and is likely to reflect chemical concentrations similar to the systemic circulation compared to our noninvasive skin surface collection which likely reflects a distinct compartment. While multiple studies have recapitulated the ability to detect drugs from skin swabs, the majority of these studies rely upon anecdotal observations without detailed clinical information, such as dosage or timing. Diphenhydramine was chosen as a probe in this study as it had been previously detected from skin swabs of transplant patients, and the safety profile was favorable in providing it to healthy volunteers [10]. We report the clinical study of diphenhydramine to measure the observation and timing of orally administered drugs via concurrent plasma and skin samples (collected at 3 regions).

## Materials and methods

### Study cohort details

We conducted a prospective, single centered, controlled study at the UCSD Altman Clinical and Translational Research Institute that was approved by the UCSD Human Research Protections Program (IRB #191026). Healthy participants were recruited by posted flyers throughout the UCSD Health System and main university campuses.

We excluded those with chronic diseases including hypertension, glaucoma, pregnancy, nursing and any skin conditions, abrasions, or compromised skin integrity. In addition,

participants were excluded if they had an allergy to diphenhydramine, active smokers (tobacco, marijuana, e-cigarettes, or vaping), chronic moderate alcohol use, chronic diphenhydramine use, or diphenhydramine use within seven days prior to the study. Those taking medications that were known CYP2D6 inducers or inhibitors or who were unable to comply with study requirements were also excluded.

Ten healthy human participants provided informed consent. The cohort consisted of 50% male, mean (SD) age 26.3 (9.41), 50% Asian, 20% African-American, 20% Hispanic, and 10% White. There were no serious adverse events with some subjects experiencing mild sedation from the diphenhydramine.

On the day of the study, subjects refrained from using topical products such as lotions or creams on their face, arms, and back for accurate skin sampling. Prior to the administration of medication, baseline skin and plasma samples were taken. A single dose of oral diphenhydramine 50 mg was administered to each subject. Blood (5 mL) and skin swab samples were taken at the following time points: 0, 0.5, 1, 1.5, 2, 4, 6, 8, 10, 12, and 24 hours. Skin swabbing was performed by using pre-soaked (ethanol:water, 1:1) cotton swabs to rub the skin surface in a circular motion for 10 seconds with both sides of the swab. Moderate, painless pressure was applied in designated locations including the forehead, upper back, and outer forearm. Adverse events were recorded throughout the study. Subjects received standardized low-fat meals during the study day. Plasma and skin samples were stored at -80°C until they were analyzed.

## Untargeted LC-MS/MS

Water (Optima LC-MS grade, W64), acetonitrile (Optima LC-MS grade, A9554), methanol (HPLC grade, A4524), ethanol (Koptec's Pure Ethanol 200 Proof, V1016), and formic acid (Optima LC-MS grade, A11750) were purchased from Fisher Scientific (Houston, TX, USA). The analytical column (Kinetex C18 1.7 µm, 100 Å, 2.1 mm internal diameter by 50 mm in length), guard cartridge (SecurityGuard ULTRA Cartridge, UHPLC C18 for 2.1 mm internal diameter columns), and Phree™ Phospholipid Removal Kit (30 mg/well, 96-well plate) were purchased from Phenomenex (Torrance, CA, USA). Eppendorf® Microplate 96/U-PP (Millipore Sigma, Burlington, MA, USA); 96-well Storage Mat IIITM 3080 (Corning, Salt Lake City, UT, USA); non-sterile Zone-FreeTM Sealing Films (ZAF-PE-50) (Excel Scientific, Victorville, CA, USA); and 1.5 mL polypropylene tubes (Axygen) were used. Six-inch cotton swabs with wooden handles (806-WC) were purchased from Puritan Medical Products (Guilford, ME, USA).

Swabs were pre-cleaned in ethanol-water (1:1) by soaking swabs overnight, decanting solution, and replacing with fresh solution. The overnight soaking process was performed three consecutive times. Prior to use in sampling, the swabs were placed in ethanol-water (1:1) using only enough solution to cover the cotton tip.

Plasma was stored at -80°C prior to extraction and allowed to thaw at room temperature. The 96-well plate Phree™ Phospholipid Removal Kit was rinsed with 300 µL of MeOH (100%) and centrifuged at 500 g for 5 min. The rinsing procedure was repeated 3 times, discarding the MeOH in the laboratory hazardous waste. Plasma samples in microcentrifuge tubes were vortexed for 5 s and then 50 µL of plasma were randomly pipetted into the Phree™ Phospholipid Removal Kit. 200 µL of MeOH (100%) using multichannel pipette, aspirating and dispensing five times to mix the plasma and MeOH. A 96-well plate (Eppendorf® Microplate 96/U-PP) was placed under the Phree™ Phospholipid Removal Kit to collect the sample. The Phree™ Phospholipid Removal kit and 96-well plate to collect samples was centrifuged at 500 g for 5 min. The Phree™ Phospholipid Removal kit was discarded in the solid biohazardous waste and

the sample-containing 96-well plate was evaporated until dry (CentriVap Benchtop Vacuum Concentrator, Labconco, Kansas City, MO, USA). The 96-well plate containing the dried extract was covered (Storage Mat III™ 3080) and stored at -80˚C prior to analysis. Immediately prior to analysis, the dried extract material was resuspended in 150 μL of MeOH-water (1:1) with 2 μM sulfamethizole, sonicated for 5 min, centrifuged for 5 min at 500 g, and covered with a plate sealing film (Zone-Free™ Sealing Films).

Skin swabs were stored in 96-well (deep) plates at -80˚C prior to extraction. The samples were rearranged to fit into four 96-well plates using tweezers. 600 μL of methanol-water (1:1) was added to each well using a multichannel pipette. The plates were capped, sonicated for 5 min, and allowed to rest overnight in a 4˚C fridge (2 PM to 10 AM the following day). The swabs were removed using tweezers, rinsing in between. The sample-containing 96-well plates were then evaporated until dry via centrifugal evaporation. The samples, once dry, were capped and placed into -80˚C storage until analysis. Prior to analysis, the four 96-well plates were resuspended with 200 μL of MeOH-water (1:1) with 2 μM sulfamethizole, sonicated for 5 min, and centrifuged for 5 min. 150 μL of extract was transferred from each well into 96-well plates (Eppendorf® Microplate 96/U-PP) and covered with a plate sealing film.

Samples were analyzed using an ultra-high performance liquid chromatograph (Vanquish, Thermo) coupled with an Orbitrap mass spectrometer (QExactive, Thermo). Blood samples were analyzed and then all skin samples. Chromatography was performed using a C18 analytical column Kinetex C18 and corresponding C18 guard cartridge at 30˚C temperature. 5.0 μL of extract was injected per sample. Mobile phase composition was as follows: A, water with 0.1% formic acid (*v/v*) and B, acetonitrile with 0.1% formic acid (*v/v*). Gradient elution was performed as follows: 0.0 min, 5.0% B; 1.0 min, 5.0% B; 7.00 min 100.0% B; 9.50 min, 100.0% B; 9.60 min, 5.0% B; 11.00 min, 5.00% B. Flow rate of 0.5 mL min$^{-1}$ was held constant. Heated electrospray ionization (HESI) was performed in the positive ion mode using the following source parameters: spray voltage, 3500 V; capillary temperature, 380˚C, sheath gas, 60.00 (a. u.); auxiliary gas, 20.00 (a.u.); sweep gas, 3.00 (a.u); probe temperature, 300˚C; and S-lens RF level, 20. Positive mode data were collected using data-dependent acquisition. MS$^1$ scans were collected at 35,000 resolution from *m/z* 150 to 1500 and were performed (~7 Hz) with a maximum injection time of 100 ms, 1 microscan, and an automatic gain control target of 1x10$^6$. The top 5 most abundant precursor ions in the MS$^1$ scan were selected for fragmentation with an *m/z* isolation width of 1.5 and subsequently fragmented with stepped normalized collision energy of 20, 30, and 40. The MS$^2$ data was collected at 17,500 resolution with a maximum injection time of 100 ms, 1 microscan, and an automatic gain control target of 5x10$^5$. The aforementioned details do not fully describe all settings of the method; therefore, we have provided a copy of the method files on MassIVE (MSV000085944).

## Time versus peak area curve for diphenhydramine and metabolites

For both the targeted and untargeted data, the time versus peak area curves were generated using the R packages ggplot2 [21], dplyr [22], tidyr [23], stringr [24], gplots [25], plotrix [26]. Diphen Table 2 was used for the targeted data values and the MZmine feature quantification table was used for the untargeted values.

## Quantitative measurement of diphenhydramine in plasma

Quantitative determination of diphenhydramine in human plasma was accomplished by the use of high-performance liquid chromatography with tandem mass spectrometry detection (Agilent Series 1100 LC with MS detector API4000, ESI source). Diphenhydramine-D3 was used as an internal standard. Briefly, diphenhydramine was precipitated from 50 μL of plasma

with 100 μL of 100% ACN and 20μl of supernatant was injected directly onto a C18 reversed phase HPLC column (MacMod Ace-5, 2.1 x 150 mm). The LC mobile phase consisted of HPLC grade water with 0.1% formic acid (A) and ACN with 0.1% formic acid (B). The analytical gradient started with a flow rate of 0.3 mL/min and 3% of solvent B for the first 0.5 minutes. Over the following 6.5 minutes, the flow rate and the organic solvent content were increased to 4 mL/min and 97% solvent B, respectively. At 7.5 minutes, the system returned to starting conditions for 2.5 minutes to equilibrate for the following injection. Positive ion mode electrospray ionization was used. The MS/MS transition for diphenhydramine ($m/z$ 256 to $m/z$ 167) and diphenhydramine-D3 ($m/z$ 259 to $m/z$ 167.2) were monitored with multiple reaction monitoring.

Calibration standards were used to generate an external calibration curve in human plasma using a linear regression algorithm to plot the peak area ratio versus concentration with 1/x weighting, over the full dynamic range of analyte concentrations. The method has a dynamic range of 2–5000 ng/mL. $y = 0.00157x + 0.000513$ ($r2 = 0.9969$).

Noncompartmental pharmacokinetic analysis of diphenhydramine was performed using Phoenix version 8.1 (Pharsight, Cary NC). The area-under-the-curve (AUC) from time zero to infinity ($AUC_{0-\infty}$) was calculated as the sum of AUC from time zero to the last measurable concentration ($AUC_{0-last}$) plus the ratio of the last measurable concentration and the elimination rate constant. Oral clearance was calculated as $F^*Dose/AUC_{0-\infty}$. A log-linear trapezoidal method was used to calculate $AUC_{0-last}$. Additional PK parameters calculated include Cmax, tmax, half-life, and volume of distribution.

## Synthesis of diphenhydramine *N*-glucose

Diphenhydramine·HCl (catalog no. D3630) and acetobromo-alpha-D-glucose (catalog no. A1750) were purchased from Sigma-Aldrich. Diphenhydramine·HCl was treated with 25% w/v NaOH solution and extracted three times with dichloromethane, then the organic layer was dried with $Na_2SO_4$, filtered, and concentrated *in vacuo* to yield the free base form of diphenhydramine prior to use. Diphenhydramine *N*-glucose was synthesized using a method adapted from Zhou *et al*. (https://doi.org/10.1124/dmd.109.028712) Diphenhydramine (1.0 g, 3.92 mmol, 1 equiv) and acetobromo-alpha-D-glucose (2.4 g, 5.88 mmol, 1.5 equiv) were added to a round bottom flask equipped with magnetic stirbar. The flask was evacuated and backfilled with $N_2$, then anhydrous dichloromethane (11.75 mL) was added to the vessel and the reaction mixture was allowed to stir at room temperature for 72 h under $N_2$. After this time, the organic solvent was removed *in vacuo* to yield an offwhite crude residue. The residue was then redissolved into methanol (86 mL) and 0.5 M $Na_2CO_3$ solution (44 mL) was added, then the mixture was allowed to stir for 5 h at room temperature. Upon completion, the reaction mixture was diluted with 200mL $H_2O$ until all solids were dissolved into solution, then extracted with dichloromethane ($3 \times 300$mL) to remove unreacted starting material. The aqueous layer was collected and the pH was adjusted to 5.0 using 1 M HCl. A 1 mL aliquot of material was used for purification and loaded onto a preparative HPLC column (XBridge BEH C18 OBD, 130Å, 5uM, 10mm × 150mm) attached to an Agilent 1200 HPLC system. Diphenhydramine *N*-glucose was eluted with a mobile phase consisting of solvent A (water with 0.1% formic acid) and solvent B (acetonitrile with 0.1% formic acid), at a flow rate of 5 mL/min and monitored by UV absorbance at 215 nm. A linear gradient was used as follows: 0 min, 5% B; 15 min, 50% B; 20–25 min, 99% B, where diphenhydramine *N*-glucose eluted at 9.9 min. Collected fractions were combined and concentrated *in vacuo* to yield diphenhydramine *N*-glucose as a white solid. [1]H NMR (599 MHz, DMSO) δ 7.41–7.20 (m, 10H), 5.54 (d, *J* = 9.8 Hz, 1H), 4.55 (dd, *J* = 27.1, 9.0 Hz, 1H), 3.82–3.64 (m, 4H), 3.64–3.55 (m, 1H), 3.54–3.22 (m, 3H), 3.21–3.01 (m,

8H). $^{13}$C NMR (151 MHz, DMSO) δ 141.72, 128.57, 127.64, 126.59, 82.78, 80.41, 76.95, 70.15, 68.82, 62.13, 62.12, 60.48, 49.16, 48.62. HRMS (ESI): exact mass calculated for [M]+ requires $m/z$ 418.2224, found $m/z$ 418.2228 with a difference of 0.95ppm. MS/MS spectrum is publicly available on GNPS at: https://gnps.ucsd.edu/ProteoSAFe/gnpslibraryspectrum.jsp? SpectrumID=CCMSLIB00005877199#%7B%7D.

## Results

Diphenhydramine was quantitatively analyzed in plasma samples using a targeted LC-MS/MS assay over 24 hours (representing three half-lives), Fig 1A, in an initial characterization of the cohort. Diphenhydramine was detected in plasma at 30 minutes and reached peak intensities at 2 hours post drug administration. Calculated pharmacokinetic parameters were as follows (mean ± standard deviation): AUC$_{0-\infty}$ 1374.69 ± 548.65 ng*h/mL, CL/F 42.2 ± 18 L/h, Cmax 157.6 ± 71.2 ng/mL, Tmax 2.1 ± 0.7 hr, Vd 522.5 ± 245.7 L, half-life 8.5 ± 1.0 h. Pharmacokinetic parameters of diphenhydramine in plasma were similar to those reported in the literature [21] (S1 Table **in** S1 File).

Plasma and skin samples (10 subjects from the forehead, upper back, and forearm), collected concurrently, were analyzed by the untargeted LC-MS/MS analysis approach. Untargeted LC-MS/MS observations of diphenhydramine in plasma mirror those of the targeted LC-MS/MS assay (Fig 1B). Diphenhydramine was detected on all skin sites, but delayed compared to plasma (Fig 1B). The initial observation of diphenhydramine on the skin was detected after 1.5 hours on the forehead and after 4 hours on the upper back and forearm (Fig 1B). The apex of detection occurred at 10 hours post drug administration for the forehead and upper back skin samples. The highest observed median value for diphenhydramine detected on the forearm was at 24 hours post drug administration. All subjects had detectable diphenhydramine on all sampled skin sites at the final time point. Note, the compositional differences between blood and skin preclude relative comparisons in measured peak area; however, if assumed that the skin of various regions are relatively similar then relative comparisons can be made.

*N*-desmethyldiphenhydramine, formed by cytochrome P450 2D6 (CYP2D6)-mediated metabolism [22], was detected in plasma 30 minutes after the initial detection of diphenhydramine (Fig 1C). The apex of *N*-desmethyldiphenhydramine was 4 hours after that of diphenhydramine. *N*-desmethyldiphenhydramine was detected on all three skin sites with the initial observation being delayed relative to initial detection in plasma. The initial and peak of *N*-desmethyldiphenhydramine in the majority of the subject's skin were seen first in the forehead, then upper back, and lastly on the forearm.

Diphenhydramine *N*-oxide (Fig 1D) levels in plasma reached an apex at the same time as parent diphenhydramine. Diphenhydramine *N*-oxide was observed on all three skin sites, but delayed relative to plasma. The timing of diphenhydramine *N*-oxide was the same for forehead and forearm skin samples. The peak areas for upper back were greater than those of forearm samples, and much greater than that obtained from forearm samples.

Diphenhydramine *N*-glucuronide ($m/z$ 432.202) was observed in plasma samples, but peaked approximately 2 hours later than diphenhydramine (Fig 1E). Diphenhydramine *N*-glucuronide was not detected in any of the skin samples. While verifying the presence of diphenhydramine *N*-glucuronide in plasma samples, a co-eluting ion of $m/z$ 418.222, differing from diphenhydramine *N*-glucuronide by Δ13.978, was noted. The feature was observed in blood and displayed a time versus peak area curve resembling that of diphenhydramine (Fig 1F) and sooner than diphenhydramine *N*-glucuronide. Evaluation of the MS/MS data obtained from the precursor ion, $m/z$ 418.222, (Fig 2C) revealed the characteristic fragment of $m/z$ 167.0850

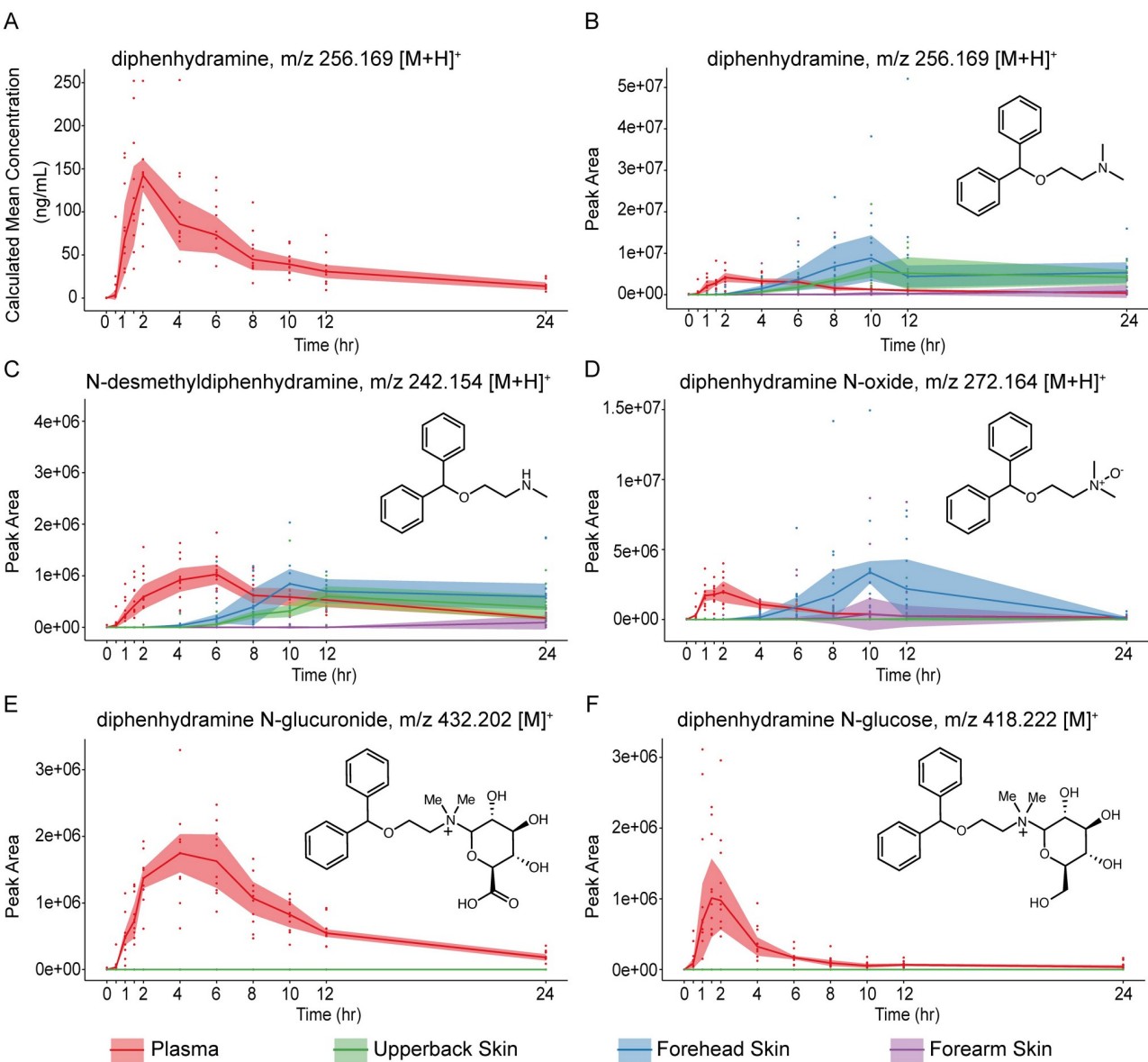

**Fig 1. Diphenhydramine and diphenhydramine metabolites observed in plasma and from skin swabs.** (**A**) quantitative measurement via targeted assay and (**B-F**) untargeted measurement of time versus peak area: (**B**) diphenhydramine, (**C**) *N*-desmethyldiphenhydramine, (**D**) diphenhydramine *N*-oxide, (**E**) diphenhydramine *N*-glucuronide, (**F**) diphenhydramine *N*-glucose. The highlighted portion of these plots represent the interquartile range for each sample type and the solid line represents the median values.

corresponding to a diphenyl carbocation (also present in diphenhydramine's MS/MS spectrum). Further, spectral evidence (accurate mass) indicated a molecular formula of $C_{23}H_{32}NO_6^+$ and the presence of a covalently bonded hexose sugar (neutral loss in the MS/MS spectrum). To confirm our putative assignment of a diphenhydramine *N*-hexose metabolite, we synthesized a chemical standard, specifically diphenhydramine *N*-glucose. Chemical synthesis was carried out using a method adapted from Zhou *et al.* [23]. We affirmed the presence of the putative hexose metabolite in our samples by additional LC-MS/MS analysis (eluting at the same retention time and same measured monoisotopic mass, isotope distribution, and MS/MS)—(Fig 2A–2C). Diphenhydramine *N*-glucose was not detected in any of the skin

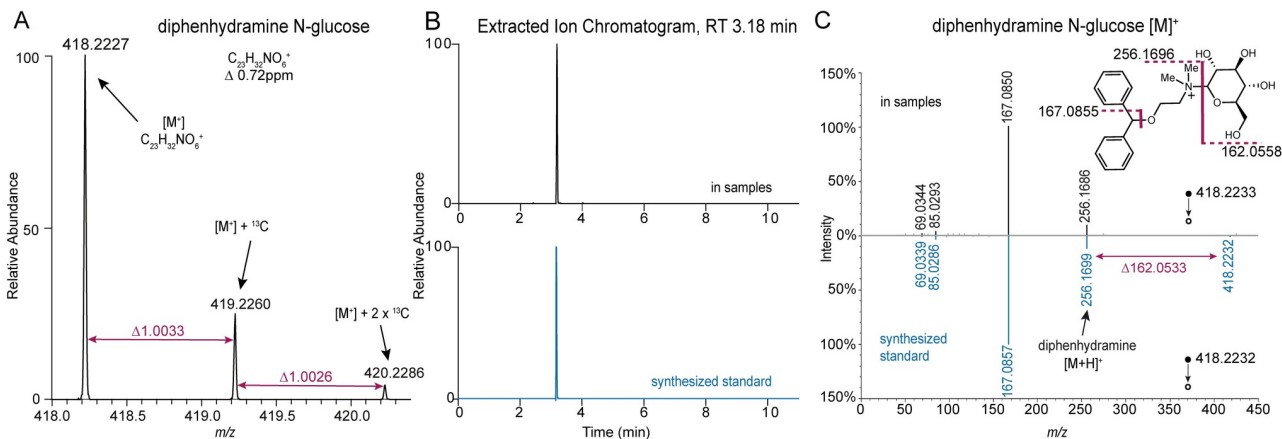

**Fig 2. Annotation of diphenhydramine *N*-glucose.** (**A**) Extracted ion chromatogram for measured and synthesized standard. (**B**) MS1 spectrum with exact mass and isotopic ratio annotation. (**C**) MS/MS measured and synthesized standard with characteristic peaks annotated.

samples. Note, the methodology used cannot unambiguously determine the stereochemistry of the hexose sugar modification in the samples; therefore, one or more potential stereoisomers may be present.

## Discussion

We report the first data on the time-course of an orally administered compound and its metabolites using untargeted mass spectrometry via non-invasively collected skin samples with relation to plasma. The detection of diphenhydramine and metabolites on the skin surface was delayed compared to plasma. The initial detection was on the forehead followed by the upper back; detection on the forearm skin was limited. We observed that the timing of detection on the surface of the skin differed by skin site (S1 Fig) and by chemical (S2 Fig). One potential explanation of the observed difference between skin sites is the variation of sebum production across the skin and between individuals [24]. Sebaceous glands are more prevalent on the forehead, and, therefore, the initial detection of diphenhydramine and metabolites on the forehead may be due to sebum secreted to the skin surface in those locations observed [25].

The ability to detect diphenhydramine, its metabolites, and other systemically administered compounds on the skin is likely impacted by their physical and chemical properties (*e.g.* lipophilicity or hydrophilicity) which is beyond the scope of this study. A recent investigation was undertaken to model and explain the previous observed drugs from skin samples. The method used multiple chemical properties to predict the likeliness of a chemical's appearance on skin with 70% accuracy [26]. In this study, we empirically determined that diphenhydramine *N*-glucuronide and diphenhydramine *N*-glucose are two metabolites that were not detected in skin samples. This concurred with the model from Bittremieux et al. as well as the agreement between the prediction and observation of diphenhydramine in skin samples. One possible explanation is that these two metabolites are heavier in mass and bulker in structure compared to diphenhydramine and the other metabolites.

Using untargeted LC-MS/MS we observed the known *N*-glucuronide metabolites and identified a previously unreported *N*-hexose metabolite. Interestingly, the N-glucose peaked earlier compared to the N-glucuronide in plasma. We hypothesize that the modification of the N-glucose may occur in the intestine as opposed to the liver. The glucuronidation pathway is a well characterized pathway performed by uridine diphosphate (UDP)-glucuronosyltransferases

(UGTs) primarily in the liver but also in the epithelia of our gut [27]. Glucuronidation of diphenhydramine appears to be performed by UGT2B10 [28] which exists primarily in the liver with little to no expression in the intestine [29]. It is possible that glucose modification occurs in the gut where trillions of microbiota exist, making diphenhydramine *N*-glucose a microbially modified drug metabolite. The direct metabolism of drugs by microbes is well documented, but not specifically diphenhydramine or hexose modification [30].

In conclusion, diphenhydramine and two known metabolites were detected on the skin over time using noninvasive skin swabs and untargeted LC-MS/MS. The timing was always delayed compared to levels observed in plasma. Untargeted LC-MS/MS was used to identify an unreported *N*-hexose metabolite which was only observed in plasma, similar to that of the known *N*-glucuronide metabolite. Our study is the first to show the time-course and relationship between plasma and the skin surface of an orally administered compound and its metabolites. This potential form of noninvasive skin detection has broad clinical applications including therapeutic drug monitoring, drug adherence, and disease state monitoring, with the purpose of providing essential clinical information in a non-invasive manner.

## Supporting information

**S1 File.**
(DOCX)

**S1 Fig. Visualization of diphenydarmine and annotated metabolites on an androgynous model displaying median values over time.** This 3D illustrative molecular map uses a white-blue color scale representative of increasing metabolite intensity for each skin site observed for each metabolite (A) diphenhydramine, (B) *N*-desmethyldiphenhydramine and (C) diphenhydramine *N*-oxide.
(TIF)

**S2 Fig. Time versus peak area of diphenhydramine and diphenhydramine metabolites observed by sample type** Plot of time vs. peak area for diphenhydramine, *N*-desmethyldiphenhydramine, diphenhydramine *N*-oxide and diphenhydramine *N*-glucuronide in (**A**) plasma, (**B**) forehead skin, (**C**) forearm skin, and (**D**) upper back skin. The highlighted portion of these plots represent the interquartile range for each sample type and the solid line represents the median values.
(TIF)

**S3 Fig. $^1$H NMR of synthesized diphenhydramine-glucose metabolite.**
(TIF)

**S4 Fig. $^{13}$C NMR of synthesized diphenhydramine-glucose metabolite in DMSO-d$_6$.**
(TIF)

## Author Contributions

**Conceptualization:** Alan K. Jarmusch, Shirley M. Tsunoda.

**Data curation:** Morgan Panitchpakdi, Emily C. Gentry, Arianna Choi, Shaden Aguirre, Kunyang Sun.

**Formal analysis:** Morgan Panitchpakdi, Kelly C. Weldon, Alan K. Jarmusch, Emily C. Gentry, Shirley M. Tsunoda.

**Funding acquisition:** Pieter C. Dorrestein.

**Investigation:** Morgan Panitchpakdi, Shirley M. Tsunoda.

**Methodology:** Kelly C. Weldon, Arianna Choi, Yadira Sepulveda, Jeremiah D. Momper, Pieter C. Dorrestein, Shirley M. Tsunoda.

**Resources:** Pieter C. Dorrestein.

**Writing – original draft:** Morgan Panitchpakdi, Kelly C. Weldon, Alan K. Jarmusch, Shirley M. Tsunoda.

**Writing – review & editing:** Morgan Panitchpakdi, Kelly C. Weldon, Alan K. Jarmusch, Emily C. Gentry, Arianna Choi, Shirley M. Tsunoda.

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
