## [Decision Letter · Decision Letter 0]

30 Mar 2022

PONE-D-21-40356Non-Invasive Skin Sampling Detects Systemically Administered Drugs in HumansPLOS ONE

Dear Dr. Tsunoda,

Thank you for submitting your manuscript to PLOS ONE. After careful consideration, we feel that it has merit but does not fully meet PLOS ONE’s publication criteria as it currently stands. Therefore, we invite you to submit a revised version of the manuscript that addresses the points raised during the review process.

The comments from both reviewers will help to clarify key parts of your work and I hope that you carefully consider each point and provide responses to improve the publication.   

We look forward to receiving your revised manuscript.

Kind regards,

Timothy J Garrett, PhD

Academic Editor

PLOS ONE

Journal Requirements:

“This research was partially supported by the National Institutes of Health, Grant UL1TR001442; the Intramural Research Program of National Institute of Environmental Health Sciences of the NIH (ES103363-01, Jarmusch); UCSD Academic Senate Grant (Tsunoda); and NIH R01 GM107550 (Dorrestein).”

We note that you have provided additional information within the Funding Section that is not currently declared in your Funding Statement. Please note that funding information should not appear in the Funding section or other areas of your manuscript. We will only publish funding information present in the Funding Statement section of the online submission form.

“This research was partially supported by the National Institutes of Health, Grant UL1TR001442; the Intramural Research Program of National Institute of Environmental Health Sciences of the NIH (ES103363-01, AKJ); UCSD Academic Senate Grant (SMT); and NIH R01 GM107550 (PCD)”

“This research was partially supported by the National Institutes of Health, Grant UL1TR001442; the Intramural Research Program of National Institute of Environmental Health Sciences of the NIH (ES103363-01, AKJ); UCSD Academic Senate Grant (SMT); and NIH R01 GM107550 (PCD)”

Reviewers' comments:

Reviewer's Responses to Questions

**Comments to the Author**

1. Is the manuscript technically sound, and do the data support the conclusions?

Reviewer #1: Partly

Reviewer #2: Yes

2. Has the statistical analysis been performed appropriately and rigorously? 

Reviewer #1: I Don't Know

Reviewer #2: Yes

3. Have the authors made all data underlying the findings in their manuscript fully available?

Reviewer #1: Yes

Reviewer #2: Yes

4. Is the manuscript presented in an intelligible fashion and written in standard English?

Reviewer #1: Yes

Reviewer #2: Yes

5. Review Comments to the Author

Reviewer #1: Panitchpakdi et al. report detection of diphenhydramine and its metabolites on skin. They take advantage of a well-established swab sampling method and untargeted mass spectrometric detection. The protocol seems to be detailed sufficiently. The report may be of interest to some other scientists focused on drug metabolism and pharmacokinetics.

Comments:

- How do we know if the amounts of skin excretions collected by skin swabbing are always the same? Moreover, sweating rates may vary among the subjects and over time. Does this variability affect the results?

- The mean age of participants (26.3) is rather low. How does it affect the results?

- Can the peak areas in Figure 1 be converted to concentrations of these analytes in sweat?

- The report does not mention some relevant previous reports and reviews, e.g. J. Mass Spectrom. 2015, 50, 1321; JALM 2020, 5, 877; Trends Endocrinol. Metab. 2021, 32, 66.

- Reference 4 is a review on hydrogel-based devices for biomedical applications. Based on its content, it is not justified to cite it after the words “and hydrogel micro patches”. Only reference 5 (or other relevant papers) should be cited in this place.

- The authors point out the limitations of the other skin sampling methods (“but these are limited to testing topical, hydrophilic substances secreted with sweat”). There is no evidence that the sampling method used in this study performs better because there is no systematic comparison.

- “co-eluting ion of m/z 418.222”: Is this species in ionic form already in the column? Or, does it become ion in the ion source, i.e. after being eluted from the column?

- Some of the text/numbers in Figure 2 are too small to read.

- Some lines in Figure 2 are so thin that they are barely visible.

Reviewer #2: This article describes the use of liquid chromatography coupled to tandem mass spectrometry (LC-MS/MS) for the analysis of swabbed skin samples to detect orally administered diphenhydramine and its metabolites. The article is brief, but well written and includes a nicely designed study combining human plasma measurements as well as non-invasive skin sampling. I only have minor comments and questions that I hope will be of use:

Specific Comments:

1. What is the clinical relevance/usage of diphenhydramine?

2. References to studies examining exogenous compounds from latent fingerprints are relevant to the current work and should be cited, including:

a. Hinners, O’Neil, Lee, “Revealing Individual Lifestyles through Mass Spectrometry Imaging of Chemical Compounds in Fingerprints,” Scientific Reports, 2018, 8, 5149.

b. Bailey, et al. “Rapid detection of cocaine, benzoylecgonine and methylecgonine in fingerprints using surface mass spectrometry,” Analyst, 2015, 140, 6254-6259.

c. Guinan, Vedova, Kobus, Voelcker, “Mass spectrometry imaging of fingerprint sweat on nanostructured silicon,” Chemical Communications, 2015, 51, 6088–6091.

d. Lauzon, Dufresne, Chauhan, Chaurand, “Development of laser desorption imaging mass spectrometry methods to investigate the molecular composition of latent fingermarks,” Journal of the American Society of Mass Spectrometry, 2015, 26, 878–886.

e. Groeneveld, de Puit, Bleay, Bradshaw, Francese, “Detection and mapping of illicit drugs and their metabolites in fingermarks by MALDI MS and compatibility with forensic techniques,” Scientific Reports, 2015, 5, 11716.

f. Kaplan-Sandquist, LeBeau, Miller, “Chemical analysis of pharmaceuticals and explosives in fingermarks using matrix-assisted laser desorption ionization/time-of-flight mass spectrometry,” Forensic Science International, 2014, 235, 68–77.

3. The m/z values at which the mass resolving powers (i.e., 35,000 and 17,500) are measured should be reported.

4. The figures are a bit blurry and the data difficult to read.

5. “Further, spectral evidence (exact mass) indicated…” should read “Further, spectral evidence (accurate mass) indicated…” (i.e., exact mass is theoretical, accurate mass refers to a measured value).

6. PLOS authors have the option to publish the peer review history of their article (what does this mean?). If published, this will include your full peer review and any attached files.

Reviewer #1: No

Reviewer #2: No

---

## [Author Response · Author response to Decision Letter 0]

17 May 2022

We have reviewed the style requirements and believe we have now adhered to your requirements.

We fully agree with data transparency and have made all of our data available publicly as noted in the following paragraph in the manuscript:

Data Processing and Data Availability

MS data, QExactive files (.raw), were converted to .mzXML files via MSConvert. Feature detection was performed using MZmine2 to produce a feature table containing detected MS1 features with associated peak area, retention time and feature ID number, parameters used can be found in Supplemental Information. All QExactive files (.raw) and .mzXML files, along with method files can be found at MassIVE (https://massive.ucsd.edu/), dataset MSV000085944. MS2 fragmentation was analyzed using the GNPS platform. https://gnps.ucsd.edu/ProteoSAFe/status.jsp?task=deee382b163f4441afea5fda4b2a2bcf

“This research was partially supported by the National Institutes of Health, Grant UL1TR001442; the Intramural Research Program of National Institute of Environmental Health Sciences of the NIH (ES103363-01, Jarmusch); UCSD Academic Senate Grant (Tsunoda); and NIH R01 GM107550 (Dorrestein).”

We note that you have provided additional information within the Funding Section that is not currently declared in your Funding Statement. Please note that funding information should not appear in the Funding section or other areas of your manuscript. We will only publish funding information present in the Funding Statement section of the online submission form.

“This research was partially supported by the National Institutes of Health, Grant UL1TR001442; the Intramural Research Program of National Institute of Environmental Health Sciences of the NIH (ES103363-01, AKJ); UCSD Academic Senate Grant (SMT); and NIH R01 GM107550 (PCD)”

“This research was partially supported by the National Institutes of Health, Grant UL1TR001442; the Intramural Research Program of National Institute of Environmental Health Sciences of the NIH (ES103363-01, AKJ); UCSD Academic Senate Grant (SMT); and NIH R01 GM107550 (PCD)”

We have placed our amended funding statement including role of funders in the cover letter as you requested.

We have provided the appropriate captions and matching in-text citations for the SI.

Reviewers' comments:

Reviewer's Responses to Questions

Comments to the Author

1. Is the manuscript technically sound, and do the data support the conclusions?

Reviewer #1: Partly

Reviewer #2: Yes

2. Has the statistical analysis been performed appropriately and rigorously?

Reviewer #1: I Don't Know

Reviewer #2: Yes

3. Have the authors made all data underlying the findings in their manuscript fully available?

Reviewer #1: Yes

Reviewer #2: Yes

4. Is the manuscript presented in an intelligible fashion and written in standard English?

Reviewer #1: Yes

Reviewer #2: Yes

5. Review Comments to the Author

Reviewer #1: Panitchpakdi et al. report detection of diphenhydramine and its metabolites on skin. They take advantage of a well-established swab sampling method and untargeted mass spectrometric detection. The protocol seems to be detailed sufficiently. The report may be of interest to some other scientists focused on drug metabolism and pharmacokinetics.

Comments:

- How do we know if the amounts of skin excretions collected by skin swabbing are always the same? Moreover, sweating rates may vary among the subjects and over time. Does this variability affect the results?

Thank you for these points that are relevant to the future of this work. Individuals swabbing the skin were instructed to swab for 10 seconds in a circular motion to maintain consistency; however, we are unable to account for varying amounts of skin excretion and sweat variability between individuals. Variability between excretion and sweat may affect these results and likely additional factors such as dehydration and thickness of the skin will require additional studies to optimize analyte detection on skin. 

- The mean age of participants (26.3) is rather low. How does it affect the results? 

This is an excellent question and according to the literature (1-5 listed below) there can be variation in sweating and skin excretions based on age, this suggests that the results could vary some between age groups. The emphasis of this study was to see if diphenhydramine could be detected on the skin over time, for future studies a larger cohort with a wider age range of participants will ideally help us to understand analyte detection variance between different age groups.

1. Inoue Y, Shibasaki M. Regional differences in age-related decrements of the cutaneous vascular and sweating responses to passive heating. Eur J Appl Physiol Occup Physiol. 1996;74(1-2):78-84. doi: 10.1007/BF00376498. PMID: 8891504.

2. Inoue Y, Nakao M, Araki T, Murakami H. Regional differences in the sweating responses of older and younger men. J Appl Physiol (1985). 1991 Dec;71(6):2453-9. doi: 10.1152/jappl.1991.71.6.2453. PMID: 1778946.

3. Foster KG, Ellis FP, Doré C, Exton-Smith AN, Weiner JS. Sweat responses in the aged. Age Ageing. 1976 May;5(2):91-101. doi: 10.1093/ageing/5.2.91. PMID: 1274803.

4. Ezure T, Amano S, Matsuzaki K. Aging-related shift of eccrine sweat glands toward the skin surface due to tangling and rotation of the secretory ducts revealed by digital 3D skin reconstruction. Skin Res Technol. 2021 Jul;27(4):569-575. doi: 10.1111/srt.12985. Epub 2021 Feb 12. PMID: 33576542; PMCID: PMC8359204.

5. Smith CJ, Alexander LM, Kenney WL. Nonuniform, age-related decrements in regional sweating and skin blood flow. Am J Physiol Regul Integr Comp Physiol. 2013 Oct 15;305(8):R877-85. doi: 10.1152/ajpregu.00290.2013. Epub 2013 Aug 7. PMID: 23926135; PMCID: PMC3798768.

- Can the peak areas in Figure 1 be converted to concentrations of these analytes in sweat?

Provided that sweat volume was not accounted for in this study the peak areas represented in Figure 1 are only able to be representative of relative abundance. Obtaining concentrations and fine tuning the sampling method for detecting and quantitating analytes on the skin is the intended direction of future studies.

- The report does not mention some relevant previous reports and reviews, e.g. J. Mass Spectrom. 2015, 50, 1321; JALM 2020, 5, 877; Trends Endocrinol. Metab. 2021, 32, 66. 

Thank you for these relevant references, J. Mass Spectrom. 2015, 50, 1321 and JALM 2020, 5, 877 are now mentioned in the introduction of the manuscript. As shown here “While some forms of skin monitoring exist, for example, sweat testing for cystic fibrosis, (2) sweat patches such as PharmCheck(3) and hydrogel micro patches (4,4-6) have been utilized, but these are specific to testing topical, hydrophilic substances secreted with sweat.” See line 72 page 2

- Reference 4 is a review on hydrogel-based devices for biomedical applications. Based on its content, it is not justified to cite it after the words “and hydrogel micro patches”. Only reference 5 (or other relevant papers) should be cited in this place.

Reference 4 “Deligkaris, K. et al. 2010. Hydrogel-Based Devices for Biomedical Applications. Sensors and Actuators. B, Chemical 147 (2): 765–74.” has been removed from the manuscript.

- The authors point out the limitations of the other skin sampling methods (“but these are limited to testing topical, hydrophilic substances secreted with sweat”). There is no evidence that the sampling method used in this study performs better because there is no systematic comparison.

Excellent point, many skin sampling methods are very novel and therefore comparisons between our methodology and other sampling methods that specifically analyze sweat are challenging. The referencing in the particular sentence mentioned above has been modified to reflect that skin sampling methods exist but are different from the methodology we have used in our study.

- “co-eluting ion of m/z 418.222”: Is this species in ionic form already in the column? Or, does it become ion in the ion source, i.e. after being eluted from the column?

We synthesized the ion in pure form with m/z of 418.222 and that co-eluted with our samples, indicating that it was in the ionic form when it was in the column. If this species were an in-source fragment it would not have the same retention time as the pure synthesized standard.

- Some of the text/numbers in Figure 2 are too small to read. This figure has been updated to improve legibility.

- Some lines in Figure 2 are so thin that they are barely visible. This figure has been updated to improve legibility. 

Reviewer #2: This article describes the use of liquid chromatography coupled to tandem mass spectrometry (LC-MS/MS) for the analysis of swabbed skin samples to detect orally administered diphenhydramine and its metabolites. The article is brief, but well written and includes a nicely designed study combining human plasma measurements as well as non-invasive skin sampling. I only have minor comments and questions that I hope will be of use:

Specific Comments:

1. What is the clinical relevance/usage of diphenhydramine?

Diphenhydramine is a widely-used over-the-counter antihistamine medication utilized for allergies and hypersensitivity conditions. We utilized diphenhydramine in this study because we had seen it before in previous skin studies, it is non-toxic, and widely available.

2. References to studies examining exogenous compounds from latent fingerprints are relevant to the current work and should be cited, including:

Thank you for these references. These fit within the context of our introduction section and have been added in as follows “Additional studies have demonstrated individuals fingerprints as a site of detection for a wide range of chemical compounds.(13-19)” See line 92 page 2, along with one additional reference 

17. Ismail M, Costa C, Longman K, Chambers MA, Menzies S, Bailey MJ. Potential to Use Fingerprints for Monitoring Therapeutic Levels of Isoniazid and Treatment Adherence. ACS Omega [Internet]. 2022 Apr 21; Available from: https://doi.org/10.1021/acsomega.2c01257 that was recently published after our first submission (line 387 page 9 - reference 17).

a. Hinners, O’Neil, Lee, “Revealing Individual Lifestyles through Mass Spectrometry Imaging of Chemical Compounds in Fingerprints,” Scientific Reports, 2018, 8, 5149.

b. Bailey, et al. “Rapid detection of cocaine, benzoylecgonine and methylecgonine in fingerprints using surface mass spectrometry,” Analyst, 2015, 140, 6254-6259.

c. Guinan, Vedova, Kobus, Voelcker, “Mass spectrometry imaging of fingerprint sweat on nanostructured silicon,” Chemical Communications, 2015, 51, 6088–6091.

d. Lauzon, Dufresne, Chauhan, Chaurand, “Development of laser desorption imaging mass spectrometry methods to investigate the molecular composition of latent fingermarks,” Journal of the American Society of Mass Spectrometry, 2015, 26, 878–886.

e. Groeneveld, de Puit, Bleay, Bradshaw, Francese, “Detection and mapping of illicit drugs and their metabolites in fingermarks by MALDI MS and compatibility with forensic techniques,” Scientific Reports, 2015, 5, 11716.

f. Kaplan-Sandquist, LeBeau, Miller, “Chemical analysis of pharmaceuticals and explosives in fingermarks using matrix-assisted laser desorption ionization/time-of-flight mass spectrometry,” Forensic Science International, 2014, 235, 68–77.

3. The m/z values at which the mass resolving powers (i.e., 35,000 and 17,500) are measured should be reported.

The resolving powers are described in the methods section where MS1 scans were collected at 35,000 resolution and 17,500 resolution for MS2 scans. See line 185 page 4.

“MS1 scans were collected at 35,000 resolution from m/z 150 to 1500 and were performed (~7 Hz) with a maximum injection time of 100 ms, 1 microscan, and an automatic gain control target of 1x106. The top 5 most abundant precursor ions in the MS1 scan were selected for fragmentation with an m/z isolation width of 1.5 and subsequently fragmented with stepped normalized collision energy of 20, 30, and 40. The MS2 data was collected at 17,500 resolution with a maximum injection time of 100 ms, 1 microscan, and an automatic gain control target of 5x105.”

4. The figures are a bit blurry and the data difficult to read.

Text and resolution of figures have been adjusted to make figures more legible and clear. 

5. “Further, spectral evidence (exact mass) indicated…” should read “Further, spectral evidence (accurate mass) indicated…” (i.e., exact mass is theoretical, accurate mass refers to a measured value).

Exact mass is now reflected as accurate mass in the manuscript see line 299 page 7.

“Further, spectral evidence (accurate mass) indicated a molecular formula of C23H32NO6+ and the presence of a covalently bonded hexose sugar (neutral loss in the MS/MS spectrum).”

6. PLOS authors have the option to publish the peer review history of their article (what does this mean?). If published, this will include your full peer review and any attached files.

Do you want your identity to be public for this peer review? For information about this choice, including consent withdrawal, please see our Privacy Policy.

Reviewer #1: No

Reviewer #2: No

---

## [Decision Letter · Decision Letter 1]

8 Jul 2022

Non-Invasive Skin Sampling Detects Systemically Administered Drugs in Humans

PONE-D-21-40356R1

Dear Dr. Tsunoda,

We’re pleased to inform you that your manuscript has been judged scientifically suitable for publication and will be formally accepted for publication once it meets all outstanding technical requirements.

Kind regards,

Timothy J Garrett, PhD

Academic Editor

PLOS ONE

Additional Editor Comments (optional):

Reviewers' comments:

Reviewer's Responses to Questions

**Comments to the Author**

1. If the authors have adequately addressed your comments raised in a previous round of review and you feel that this manuscript is now acceptable for publication, you may indicate that here to bypass the “Comments to the Author” section, enter your conflict of interest statement in the “Confidential to Editor” section, and submit your "Accept" recommendation.

Reviewer #2: (No Response)

2. Is the manuscript technically sound, and do the data support the conclusions?

Reviewer #2: Yes

3. Has the statistical analysis been performed appropriately and rigorously? 

Reviewer #2: Yes

4. Have the authors made all data underlying the findings in their manuscript fully available?

Reviewer #2: Yes

5. Is the manuscript presented in an intelligible fashion and written in standard English?

Reviewer #2: Yes

6. Review Comments to the Author

Reviewer #2: Most of my previous comments have been addressed. The only piece of information to add to the manuscript is the m/z value at which the mass resolving powers (i.e., 35,000 and 17,500) are measured. Resolving power is defined as m/Δm, so the m (i.e., m/z) of the resolving power is an important figure to report (along with how Δm is measured), especially in instruments for which the resolving power varies over the mass range. On Thermo instrument, the resolving power reported in the software is usually defined at m/z 200 and Δm is measured as full-width at half maximum (FWHM).

7. PLOS authors have the option to publish the peer review history of their article (what does this mean?). If published, this will include your full peer review and any attached files.

Reviewer #2: No

---

## [Editor Report · Acceptance letter]

15 Jul 2022

PONE-D-21-40356R1 

Non-Invasive Skin Sampling Detects Systemically Administered Drugs in Humans 

Dear Dr. Tsunoda:

I'm pleased to inform you that your manuscript has been deemed suitable for publication in PLOS ONE. Congratulations! Your manuscript is now with our production department. 

Kind regards, 

on behalf of

Dr. Timothy J Garrett 

Academic Editor

PLOS ONE